# The Effect of Short Treatment with Nigella Sativa on Symptoms, the Cluster of Differentiation (CD) Profile, and Inflammatory Markers in Mild COVID-19 Patients: A Randomized, Double-Blind Controlled Trial

**DOI:** 10.3390/ijerph191811798

**Published:** 2022-09-19

**Authors:** Khalid A. Bin Abdulrahman, Abdullah Omar Bamosa, Abdullah I. Bukhari, Intisar Ahmad Siddiqui, Mostafa A. Arafa, Ashfaq A. Mohsin, Mamdouh Faleh Althageel, Majed Owed Aljuaeed, Ibrahim Mohammed Aldeailej, Abdulaziz Ibrahim Alrajeh, Kamel Mohamed Aldosari, Najat Ahmed Hawsawi, Khalid Ibrahim Zawbaee, Saad Mohammed Alsurayea

**Affiliations:** 1Department of Medical Education, College of Medicine, Imam Mohammad Ibn Saud Islamic University (IMSIU), Riyadh 13317, Saudi Arabia; 2Department of Physiology, College of Medicine, Imam Abdulrahman Bin Faisal University, Dammam 34212, Saudi Arabia; 3Department of Medicine, Division of Infectious Diseases, College of Medicine, Imam Mohammad Ibn Saud Islamic University (IMSIU), Riyadh 13317, Saudi Arabia; 4Department of Dental Education, College of Dentistry, Imam Abdulrahman Bin Faisal University, Dammam 31441, Saudi Arabia; 5College of Medicine, King Saud University, P.O. Box 2925, Riyadh 11461, Saudi Arabia; 6Epidemiology Department, High Institute of Public Health, Alexandria University, Alexandria 21561, Egypt; 7Department of Pharmaceutics, College of Clinical Pharmacy, Imam Abdulrahman Bin Faisal University, Dammam 34212, Saudi Arabia; 8Ministry of Health, King Salman Bin Abdulaziz Hospital, Riyadh 12769, Saudi Arabia; 9Ministry of Health, Riyadh Regional Lab and Blood Bank, Riyadh 12746, Saudi Arabia

**Keywords:** mild COVID-19, *Nigella sativa*, anti-inflammatory, randomized clinical trial, Saudi Arabia

## Abstract

The current study investigated the impact of different doses of *Nigella sativa* seeds on the symptoms, the cluster of differentiation profile group, and inflammatory markers of mild COVID-19 cases. Methods: The study was a double-blind placebo-controlled clinical trial. Patients with mild and asymptomatic SARS-CoV-2 infection patients were randomly subdivided into seven subgroups: Group (GP) 1: received charcoal capsules as a control group, and GP 2: received three capsules of whole *Nigella sativa* seeds daily, two capsules in the morning and one in the evening; GP 3: received three capsules of whole *Nigella sativa* seeds every 12 h, GP 4: received five capsules in the morning and four capsules of whole *Nigella sativa* seeds in the evening, GP 5: received one capsule of *Nigella sativa* powder every 12 h; GP 6: received two capsules of *Nigella sativa* powder every 12 h; GP 7: received three capsules of *Nigella sativa* powder every 12 h; all treatment course was for ten days. Inflammatory parameters were assessed before and after interventions. Results: 262 subjects were included in the final analysis. No significant difference was detected regarding age, gender, and nationality. No significant differences were detected between the inflammatory marker in all groups. The WBCs showed a significant difference between before and after the intervention. While for procalcitonin, a significant difference was demonstrated in groups 1,4, and 6. Conclusions: The current randomized clinical trial did not reveal a significant effect of ten days of treatment with various doses of *Nigella sativa* on symptoms, differentiation profile, and inflammatory markers of patients with COVID-19. As a natural product, the effect of *Nigella sativa* on disease requires weeks to manifest itself.

## 1. Introduction

The spread of severe acute respiratory syndrome coronavirus 2 (SARS-CoV-2) has caused the world pandemic coronavirus disease-19 (COVID-19), which resulted in a strenuous global situation in all essential aspects related to human life on earth, including; survival, health care services, economic status, social relations, activities, and religious obligations. The first cases of this pandemic were reported in December 2019 in Wuhan, Hubei, China [1]. Since then, scientists around the globe have been investigating several potential drugs to cure this disease. *Nigella sativa* is one of the most used herbal products in various communities to prevent and treat many diseases. *Nigella sativa* has been reported to have antiviral, immunomodulation, antioxidant, and anti-inflammatory effects [2,3]. These beneficial effects are closely related to the pathophysiology of COVID-19, which led several investigators to suggest *Nigella sativa* as a potential treatment for SARS-CoV-2 infection [3,4,5].

*Nigella sativa* is one of the most widely used herbal treatments in the world. It possesses a wide range of therapeutic and preventive effects, including; antioxidant, antimicrobial, antidiabetic, anti-dyslipidemic, antihypertensive, anti-inflammatory, anticonvulsant, anticancer, cardioprotective, nephroprotective, neuroprotective, hepatoprotective, and immunomodulatory [6].

The antiviral effect of *Nigella sativa* against several viruses has been documented in various in vivo and in-vitro studies. *Nigella sativa* oil has produced a substantial antiviral effect against murine cytomegalovirus. The virus titer was undetectable on days 3 and 10 in the liver and spleen of mice treated with *Nigella sativa* oil while active in the controlled ones [7]. Investigators attributed this antiviral effect to an increase in the number and function and an increase in IFN-γ. Interestingly, in cell culture, *Nigella sativa* extract reduced coronavirus load to an undetectable level 6 h post-infection and 10% control at 8 h [8]. Turkeys infected with the H9N2 avian influenza virus showed a dose-dependent decrease in the virus titer as the bird’s diet was supplemented with 2%, 4%, and 6% *Nigella sativa*, and the titer was at levels of 0 on day 8 of the infection for the 6% dose of *Nigella sativa* [9]. The formulation of *Nigella sativa* (Alpha-zam) was found to induce selective inhibition of Hepatitis C virus in cell culture; The investigators reported that this anti-hepatitis C activity of *Nigella sativa* was independent of interferon [10]. A clinical trial that evaluated the effect of Nigella sativa oil on patients with hepatitis C virus reported a more than 50% decrease in virus titer after three months of taking 450 mg of *Nigella sativa* oil three times daily. The patients improved their laboratory investigations and symptoms [11]. An interesting case report showed a wonderful effect of the *Nigella sativa* concoction (60% *Nigella sativa* and 40% honey), resulting in complete remission of HIV patients after taking 10 ml of the concoction for three months. Fever and diarrhea disappeared in 7 days, while the virus titer was undetectable in the second month, and the patient became seronegative after 6 months [12].

The anti-inflammatory and immunomodulatory effects of *Nigella sativa* have been studied extensively. A randomized, double-blind, placebo-controlled clinical trial in 42 rheumatoid arthritis patients supplemented with two capsules of 500 mg *Nigella sativa* oil for eight weeks resulted in a significant elevation of the anti-inflammatory cytokine IL10 while NO decreased significantly [13]. In another randomized clinical trial, 3 g of *Nigella sativa* oil supplemented to women on a low-calorie diet for eight weeks significantly reduced tumor necrosis factor-alpha and C reactive protein compared to the placebo group [14]. The immunomodulatory effect of *Nigella sativa* is associated with increased natural killer cells and immunity of T cells [15].

Given these beneficial effects of *Nigella sativa*, the current study was conducted to investigate the impact of different doses of grinded and whole black seeds on the symptoms, CD profile, and inflammatory markers of patients with mild COVID-19.

## 2. Materials and Methods

This study was part of an interventional clinical trial aiming to examine the efficacy of black seeds in the immunity of patients with mild COVID-19 in Riyadh city, Saudi Arabia.

### 2.1. Study Design

The study was a double-blind placebo-controlled prospective, multiple-armed clinical trial. The Institutional Review Board (IRB) of Imam Mohammad Ibn Saud Islamic University has approved this study under project number 22-2020, session 32, dated 7 May 2020. The Saudi FDA registered the study as SCTR no. 20051303, dated 29 June 2020. Furthermore, the Saudi MOH had approved the RCT through IRB number 20-147E dated 20 July 2020.

### 2.2. Nigella sativa Preparation

*Nigella sativa* (*N.S.*) seeds were prepared in the form of 500 mg powder and 350 mg whole seed oral capsules. The above procedures were carried out at the pharmaceutical lab of the College of clinical pharmacy at Imam Abdulrahman Bin Faisal University, Dammam, Saudi Arabia. Charcoal capsules were used as a placebo control and placed in the same bottles as *Nigella sativa* capsules. All capsules were distributed in transparent cellulose capsules of the same size and appearance for both the test and placebo groups.

### 2.3. Determination of Sample Size

Our intervention is similar to many trials that studied the efficacy of their interventions against the SARS-CoV-2 virus; one of those studies shows that the effectiveness of Hydroxychloroquine is 70% [16]. Our study assumes that 70% of patients who receive *Nigella sativa* will be cleared of the SARS-CoV-2 virus within 7 days. Therefore, our statistical parameters are as follows; 1 *Nigella sativa* 70% will be -ve, two placebo 34% will be -ve, and three 1:1 enrollment ratios: For most studies, the enrollment ratio is 1 (that is, equal enrollment between both groups). So our enrollment is 30 for each group 4 type 1 error alpha: 0.05. The probability of a type-I error determining a difference between two groups when such a distinction does not exist (false positive rate). Most of the medical literature uses a value of 0.05. The power of the study is 80%, the ability to detect a difference between groups when there is a difference. Power is 1-β, where β is the risk of a type II error (false-negative rate). Most medical literature uses a value of 80 to 90% power (β of 0.1–0.2). Sample size calculation: 29 patients in each group [17].

### 2.4. Study Population

Subjects were recruited from health quarantines in Riyadh city, namely Al Izdihar Holiday Inn Hotel quarantine, Marriott Hotel quarantine, and home quarantines linked to AlMorouj primary health care center. The inclusion criteria were: (1) 18 up to 85 years, (2) Positive COVID-19 PCR diagnostic test for COVID-19, and (3) Not in severe respiratory distress that requires critical care and admission to the ICU. The exclusion criteria were: (1) a history of drug addiction or (2) pregnant and lactating women or (3) malignancy, or (4) chronic illnesses except for hypertension and diabetes mellitus. All patients who agreed to participate were asked to sign a written consent form. Qualified healthcare professionals conducted a detailed history and clinical examination. The patients were interviewed by telephone to complete a questionnaire prepared in Arabic and English and built on the Survey Monkey software program.

The questionnaire included personal information, history of illnesses, immunization history, immunity-related questions, body mass index, and detailed symptoms of COVID-19 infection. Two hundred and sixty-two (262) confirmed patients with mild or asymptomatic were randomly enrolled in the study. The selected dose of 1-3 g of powdered black seed is based on previous clinical trials. This range of doses covered the therapeutic range for eradicating H Pylori or improving glycemic control [18,19]. The subjects were subdivided into seven subgroups: Group 1: received one capsule of charcoal every 12 h for ten days and served as a control group; Group 2: received three capsules daily, two capsules in the morning and one in the evening (each capsule had 350 mg of whole *N.S.* seeds) every 12 h for ten days, Group 3: received three capsules (Each capsule has 350 mg of whole *N.S.* seeds) every 12 h for ten days, Group 4: received five capsules every morning and four capsules in the evening for ten days (Each capsule has 350 mg of whole *N.S.* seeds), Group 5: received one capsule of 500 mg of *N.S.* Powder every 12 h for ten days; Group 6: received two capsules of 500 mg of *N.S.* Powder every 12 h for ten days; Group 7: received three capsules of 500 mg of N.S. Powder every 12 h for ten days. The following anti-inflammatory parameters were assessed before and after interventions: ProCalcitonin (ng/mL), Ferritin (µg/L), CRP (mg/L), ESR (mm/h) and WBC (×10^3^/µL). The PCR test was performed before and after the interventions.

### 2.5. Randomization and Blindness

Randomization was achieved through a computer-generated table to distribute the 262 patients into the seven subgroups. The code for each subgroup was created by the investigator who prepared the capsules and was not involved in patient recruitment or follow-up. The code was unmasked after data analysis. The charcoal was packed in capsules identical to *Nigella sativa* and looked the same as *Nigella sativa* capsules in black color and acted as a control group. The test capsules and placebo were packaged in the same bottles, and the patients could not distinguish between them. Blinding was 100% successful, as all patients in the placebo group thought they were given *Nigella sativa* capsules. A study coordinator was responsible for interacting with the study participants (unblinded) to ensure not break the blinding. According to the updated version-2 of the Saudi Ministry of health COVID-19 treatment protocol published on 17 June 2020 [20], no interventional treatment shall be administered to asymptomatic or mild and moderate symptoms. Therefore, study subjects were not receiving antiviral treatment or any other standard of care during the RCT study period.

### 2.6. The Primary Outcome

The effect of *Nigella sativa* on signs and symptoms of mild cases.

### 2.7. The Secondary Outcome

Effect of *N.S.* on inflammatory markers and CD profile in mild and asymptomatic cases of COVID-19.

### 2.8. Statistical Analysis

Statistical data were analyzed using SPSS-20.0, an IBM product of Chicago (IL, USA). Categorical response variables were presented in frequencies and percentages, including sex, nationality, symptoms, and general patient condition before and after treatment. The Chi-square test was applied to compare categorical variables between study groups. Numeric data based on inflammatory markers were presented as mean and standard deviation and explored for the normality test using the Kolmogorov-Smirnov test specific for each group, revealing that the variables were normally distributed. Paired sample *t*-test was applied to compare pre vs. Post-test observations within-group effect. On the contrary, ANOVA evaluated the effect of post-intervention mean inflammatory markers after the intervention. A *p*-value < 0.05 was considered a statistically significant result.

## 3. Results

The number of subjects included in the final analysis was 262; Their distribution across the seven groups and the follow-up days is illustrated in Figure 1.

The CONSORT chart illustrated the number of patients included, randomized, lost to follow-up, and analysis. Although 75 patients were lost to follow-up, the main reasons for withdrawing from RCT were recovery from the symptoms before ten days of quarantine and being bothered by post-blood testing. None of the recruited patients had side effects from *N.S.* or placebo capsules.

The distribution of cases by age, sex, and nationality is shown in Table 1.

The nonsignificant difference in demographic characteristics between the groups.

Two-thirds (66%) were Saudi, and 70% were males. The mean age was 35.2 + 11.4 (18 to 71 years). The highest mean age was observed in group 6, NS-G2 (38.1 ± 12.4), and the lowest was in group 5, NS-G1 (32.6 ± 14.4). No significant differences were detected concerning age, gender, and nationality, *p* > 0.05.

The proportion of patients with fever as a significant symptom of SARS-CoV-2 infection ranged from 5% to 27.8% on day one of the trials in the seven study groups (*p* = 0.173). It was 0% on post-test day ten in all groups, which revealed a nonsignificant difference. The proportion of patients who had cough was comparatively higher than fever, ranging from 28% to 38.9% on day one of the trial in the study groups (*p* = 0.945). On day 10, a relatively lower proportion was observed in all groups (*p* = 0.869). On day 1, loss of sense of smell was the most common symptom prevalent in all study groups (23.7% to 53.1%, *p* = 0.258). It was more prevalent on day five than on day ten in some groups (*p* = 0.675) and persistent until day 10, ranging from 6% to 30.8% (*p* = 0.529), as presented in Table 2.

ANOVA was used to compare the means of the post-test group. No significant differences were detected for the eight inflammatory markers. In the seven groups of the trial, *p* > 0.05. WBC and Lymphocytes showed a significant difference between pre-and post-intervention readings. While for procalcitonin, a significant difference was demonstrated in groups 1, 4 and 6, Table 3. All PCT tests were negative after ten days of treatment.

The results of the post-test CD profile were not significantly different between the seven groups (*p* > 0.05). The percentage of CD3 T cells showed a non-significant difference between baseline and post-test in all groups. Compared to their baseline values, the mean CD19 cell rates were significantly elevated after the post-test in all groups (*p* < 0.05). On the other hand, the percentage of CD16+/CD56 T cells showed a significant decrease in all groups except group 5 (*p* = 0.136), while group 3 was marginally significant (*p* = 0.055). Comparing the mean CD3+/CD4+ T cell rates at baseline and post-test showed a nonsignificant difference in all groups except group 7 (*p* = 0.033). The elevation in absolute CD3 counts was only significant in groups 2 (*p* = 0.023) and 6 (*p* = 0.003). However, the mean absolute CD19 count was significantly increased in all groups except group 3. The absolute count of CD16+ CD56 after the test compared to baseline showed a significant decrease in group-3 (*p* = 0.028). Compared to baseline, the post-test mean of the absolute count value was significantly elevated in groups 2, 4, 6, and 7 but did not change significantly in groups 1, 3, and 5. Furthermore, compared to the baseline value, the post-test mean of CD3+/CD8+ absolute count was significantly increased in groups 1, 6, and 7 (Table 4).

## 4. Discussion

The literature lacks clinical trials measuring the anti-inflammatory effect of whole and grinded black seed in different doses for a short duration in patients with mild SARS-CoV-2 infection. The current study showed an insignificant effect of *Nigella sativa* whole and crushed between the control and study groups for the three *Nigella sativa* doses (1–3 g). There was no significant effect of *N.S.* on signs and symptoms, CD profile, and essential inflammatory markers in mild cases of Saudi patients. Our findings on signs and symptoms, CD profile, and inflammatory markers are heading in the same direction. This supports the conclusion of the insignificant effect of ten days of *Nigella sativa* on mild COVID-19 cases. The results of the current study appear to conflict with the vast literature documenting *the* antiviral and anti-inflammatory effects of Nigella Sativa [3,21,22,23]. However, the literature is mainly based on in vitro and animal studies, and clinical trials on the impact of Nigella sativa in mild cases of infectious diseases are limited. The only previous study in the literature that examined *Nigella sativa* in mild COVID-19 patients used Nigella sativa oil. It was an open-label randomized clinical trial, and the authors reported a significant impact of *Nigella sativa* oil on common signs and symptoms of COVID-19 [24]. However, the conclusion of this study should be taken with caution as the reported results are related to subjective parameters, which in such open-label studies have a significant bias. A recent randomized placebo-controlled double-blind clinical trial examined the effect of a concoction of *Nigella sativa* and honey on moderate and severe cases of COVID-19 and reported a significant impact of treatment on mortality rate and hospital stay compared to placebo [25]. Although showing a good product of *Nigella sativa*, this study does not conflict with our results as it dealt with moderate and severe cases rather than mild COVID-19 in our study.

The negative effect of *Nigella sativa* in the current study could be attributed to the short duration of the treatment course of 10 days and the subsiding nature of the mild SARS-CoV-2 infection. In such conditions of acute self-limiting viral infections, the challenge of the subsiding nature of the disease should be overcome by a larger sample size, which is a limitation of our study. Furthermore, the duration of treatment in our study was only ten days.

Clinical trials conducted on the effect of *Nigella sativa* on various chronic illnesses were of 4 to 12 weeks duration [18]. The beneficial effects of natural products usually take weeks, and the longer the treatment, the better the outcome. Treatment with *Nigella sativa* for 6–12 weeks produced significant beneficial effects on H-Pylori eradication [19], glycemic control in diabetic patients [26], kidney stone disillusions [27], and frequency of seizures in pediatric patients [28]. In a randomized placebo-controlled clinical trial, 2.5 mL of *Nigella sativa* oil administered twice daily to hypertensive patients significantly reduced systolic and diastolic blood pressures after six and eight weeks of treatment; however, the effect was insignificant at three weeks [29]. The clinical trials conducted on the antiviral effect of *Nigella sativa* were also of several weeks duration. A study conducted on 51 HIV patients treated with *Nigella sativa* and honey concoction revealed clearance of symptoms in four weeks in all patients, while CD4 increased significantly, and the viral load was undetectable in more than 80% of the patients at the end of 16 months of treatment duration [30]. Another study gave six HIV patients a *Nigella sativa* and honey concoction for 16 weeks. All symptoms disappeared at 20 days, CD4 increased significantly in all patients, while viral load decreased significantly after eight weeks and was undetectable at weeks 17 and [31]. From these studies on different diseases, we could conclude that clinical trials of *Nigella sativa* should not take less than four weeks to evaluate this natural product’s effect on the progress of the disease. Our trial was initially planned for such a duration, but we had to reduce the treatment period because the Saudi Ministry of health’s COVID-19 protocol has restricted quarantine to ten days. It cannot be confirmed that there is no clinical or laboratory anti-inflammatory impact of black seed as an antidote to the COVID-19 virus in mild cases. The duration of treatment is relatively short in the current study compared to previous studies. Additionally, the nature of respiratory viral diseases—including COVID-19—for mild cases, the symptoms are usually mild and disappear within days without therapeutic intervention.

A recently published study showed that 12% of mild cases of COVID-19 patients in Riyadh were asymptomatic. The top five clinical manifestations of mild cases of COVID-19 were general fatigue, headache, cough, anosmia, and ageusia [32]. In our study, the proportion of fever ranged from 5% to 27.8% on day 1 of the trials in the seven study groups (*p* = 0.173). However, the study revealed a nonsignificant difference between the groups. Not surprisingly, the proportion of patients who presented cough was comparatively higher than that of fever, ranging from 28% to 38.9% on day 1 of the trials was statistically consistent in the study groups (*p* = 0.945). As reported in previous studies, some patients had only a mild fever, mild fatigue, or no symptoms [33,34,35]. Loss of smell and taste were the most common symptoms prevalent in all study groups through the 10-day follow-up. A recently published systematic review and meta-analysis reported anosmia in 12,038 patients out of 32,142 patients with COVID-19 from 107 studies, with a prevalence of 38.2%. On the contrary, dysgeusia was reported in 11,337 patients of 30,901 COVID-19 patients in 101 studies, with a prevalence of 36.6% worldwide. Furthermore, the prevalence of anosmia was 10.2-fold higher, and that of dysgeusia was 8.6-fold higher in COVID-19 patients compared to those with other respiratory infections or diseases similar to COVID-19.

Our study showed a significant elevation in white blood cells and lymphocyte counts by the end of the ten-day duration of the ten days in all study groups. This elevation coincides with the disappearance of most symptoms in all groups. This finding is consistent with the typical findings of leukopenia and lymphopenia in mild cases of COVID-19 [36,37]. The significant increase in the absolute count of both CD4 and CD8 cells in most groups could be due to the development of memory cells of both types. This explanation is supported by previous reports on the development of memory cells of CD4 and CD8 memory cells in asymptomatic and mild cases of COVID-19 [38]. Our study shows a significant increase in the percentage and absolute count of CD19 (B lymphocyte antigen) in all groups at the end of the ten days of treatment. This agrees with the previous report that B lymphocyte memory cells are the major ones developed after COVID-19 infection [39].

## 5. Conclusions

The effect of *Nigella sativa*, as a natural product, on disease requires weeks to manifest itself. The current RCT did not reveal a significant effect of ten days of treatment with various doses of *Nigella sativa* on symptoms and inflammatory markers in COVID-19 patients. Therefore, it is not recommended to study the effect of an herb on subsiding self-limited diseases.

## 6. Limitations

The study has a few limitations; the course of *Nagila sativa* treatment was too short to observe any clinical and laboratory effects. Furthermore, 30 subjects in each subgroup are relatively small sample sizes to study the impact of the natural product on a self-limited disease.

## Figures and Tables

**Figure 1 ijerph-19-11798-f001:**
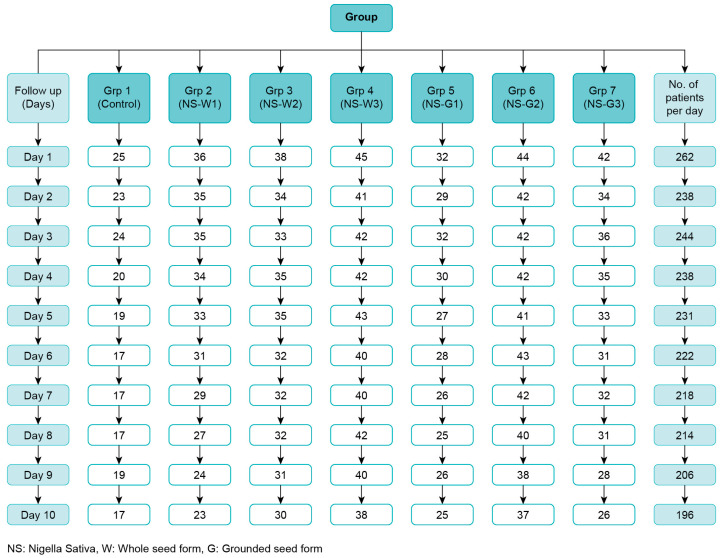
The distribution of patients in the seven groups, randomization, loss to follow-up, and analysis.

**Table 1 ijerph-19-11798-t001:** Patient demographic characteristics.

Demographic Characteristics	Total (n)	Age (Years)	Gender		Nationality	
		Mean ± SD	Male	Female	Saudi	Non-Saudi
Groups	262	35.2 ± 11.4	183	79	172	90
Group-1 (Control)	25	36.1 ± 9.6	18 (72.0)	7 (28.0)	12 (48.0)	13 (52.0)
Group-2 (NS-W1)	36	36.0 ± 9.5	28 (77.8)	8 (22.2)	20 (55.6)	16 (44.4)
Group-3 (NS-W2)	38	33.1 ± 11.3	29 (76.3)	9 (23.7)	27 (71.1)	11 (28.9)
Group-4 (NS-W3)	45	36.0 ± 11.8	29 (64.4)	16 (35.6)	28 (62.2)	17 (37.8)
Group-5 (NS-G1)	32	32.6 ± 14.4	20 (62.5)	12 (37.5)	26 (81.2)	6 (18.8)
Group-6 (NS-G2)	44	38.1 ± 12.4	30 (68.2)	14 (31.8)	29 (65.9)	15 (34.1)
Group-7 (NS-G3)	42	33.9 ± 13.2	29 (69.0)	13 (31.0)	30 (71.4)	12 (28.6)
Sig.		*p* = 0.345	*p* = 0.757		*p* = 0.130	

Values given in parentheses are percentages.

**Table 2 ijerph-19-11798-t002:** Comparison of symptoms of COVID-19.

Symptoms	Days	Group						
		I	II	III	IV	V	VI	VII
Fever	1	4 (16.0)	10 (27.8)	2 (5.3)	7 (15.6)	3 (9.4)	5 (11.4)	6 (14.3)
	5	3 (15.8)	2 (6.0)	1 (2.9)	3 (7.0)	1 (3.7)	1 (2.4)	1 (3.0)
	10	0 (0.0)	0 (0.0) ^b^	0 (0.0)	0 (0.0)	0 (0.0)	0 (0.0)	0 (0.0)
Cough	1	7 (28.0)	14 (38.9)	14 (36.8)	13 (28.9)	10 (31.2)	13 (29.5)	14 (33.3)
	5	6 (31.6)	7 (21.2)	10 (28.6)	12 (27.9)	8 (29.6)	15 (36.6)	9 (27.3)
	10	3 (17.6)	3 (13.0)	5 (16.7)	7 (18.4)	4 (16.0)	9 (24.3)	7 (26.9)
Sense of smell	1	8 (32.0)	14 (38.9)	9 (23.7)	18 (40.0)	17 (53.1)	20 (45.5)	17 (40.5)
	5	4 (21.1)	13 (39.4)	16 (45.7)	16 (37.2)	11 (40.7)	15 (36.6)	15 (45.5)
	10	1 (5.9)	4 (17.4)	5 (16.7)	8 (21.1) ^b^	7 (28.0) ^b^	8 (21.6) ^b^	8 (30.8)
Lose sense of Taste	1	6 (24.0)	11 (30.6)	9 (23.7)	15 (33.3)	17 (53.1)	20 (45.5)	15 (35.7)
	5	4 (21.1)	11 (33.3)	15 (42.9)	14 (32.6)	11 (40.7)	11 (26.8)	10 (30.3)
	10	1 (5.9) ^b^	3 (13.0)	4 (13.3)	7 (18.4)	4 (16.0) ^b^	7 (18.9) ^b^	3 (14.8)
Shortening of breath	1	3 (12.0)	2 (5.6)	3 (7.9)	5 (11.1)	3 (9.4)	4 (9.1)	4 (9.5)
	5	2 (10.5)	3 (9.1)	0 (0.0)	3 (7.0)	2 (7.4)	2 (4.9)	3 (9.1)
	10	1 (5.9)	0 (0.0)	0 (0.0)	1 (2.6)	1 (4.0)	2 (5.4)	0 (0.0)
Nasal Discharge	1	4 (16.0)	9 (25.0)	6 (15.8)	10 (22.2)	3 (9.4)	9 (20.5)	6 (14.3)
	5	6 (31.6)	4 (12.1)	8 (22.9)	5 (11.6)	4 (14.8)	6 (14.6)	4 (12.1)
	10	0 (0.0) ^b^	2 (8.7)	3 (10.0)	4 (10.5)	2 (8.0)	1 (2.7)	1 (3.8)
Sore Throat	1	1 (4.0)	8 (22.2)	7 (18.4)	10 (22.2)	5 (15.6)	5 (11.4)	3 (7.1)
	5	1 (5.3)	2 (6.1)	2 (5.7)	9 (20.9)	5 (18.5)	6 (14.6)	1 (3.0)
	10	1 (5.9)	0 (0.0) ^b^	0 (0.0)	3 (7.9)	3 (12.0)	1 (2.7)	1 (3.8)
General fatigue	1	15 (16.0)	16 (44.4)	16 (42.1)	21 (46.7)	8 (25.0)	20 (45.5)	18 (42.9)
	5	6 (31.6)	6 (18.2)	7 (20.0)	12 (27.9)	5 (18.5)	14 (34.1)	8 (24.2)
	10	1 (5.9)	2 (8.7) ^b^	0 (0.0) ^b^	3 (7.9) ^b^	1 (4.0) ^b^	2 (5.4) ^b^	3 (11.5) ^b^
Headache	1	7 (28.0)	18 (50.0)	17 (44.7)	20 (44.4)	11 (34.4)	16 (36.4)	8 (19.0)
	5	4 (21.1)	6 (18.2)	7 (20.0)	11 (25.6)	5 (18.5)	10 (24.4)	7 (21.2)
	10	1 (5.9) ^b^	1 (4.3)	2 (6.7) ^b^	3 (7.9) ^b^	4 (16.0)	1 (2.7) ^b^	4 (15.4)

Nonsignificant difference between groups on day one versus day five vs. day ten at *p* < 0.05. ^a^ Significant difference between any two groups at *p* < 0.05. ^b^ Significant difference within the group compared to day one vs. day five vs. day 10, at *p* < 0.05.

**Table 3 ijerph-19-11798-t003:** Comparison of mean levels of inflammatory markers between groups.

Inflammatory Markers		Before vs. after Treatment Tests (Mean ± S.D)							Post-Test (ANOVA)
		Group-1 (n = 19)	Group-2 (n = 22)	Group-3 (n = 26)	Group-4 (n = 40)	Group-5 (n = 20)	Group-6 (n = 34)	Group-7 (n = 26)	
ProCalcitonin (ng/mL)	Before	0.06 ± 0.04	0.05 ± 0.04	0.03 ± 0.02	0.03 ± 0.02	0.02 ± 0.00	0.04 ± 0.03	0.06 ± 0.07	F = 1.11 *p* = 0.382 *
	After	0.03 ± 0.01	0.03 ± 0.02	0.02 ± 0.01	0.02 ± 0.01	0.02 ± 0.01	0.02 ± 0.01	0.03 ± 0.02	
	Sig.	0.014 **	0.357	0.106	0.005 **	0.829	0.004 **	0.228	
Ferritin (µg/L)	Before	139.4 ± 93.5	127.8 ± 134.4	164.5 ± 170.2	104.1 ± 119.6	81.2 ± 82.6	124.7 ± 105.8	122.7 ± 153.1	F = 0.866 *p* = 0.587 *
	After	135.8 ± 105.8	153.3 ± 165.6	135.3 ± 147.5	106.4 ± 102.4	94.1 ± 119.6	150.9 ± 158.1	101.9 ± 112.1	
	Sig.	0.097	0.381	0.664	0.971	0.176	0.150	0.632	
CRP (mg/L)	Before	7.02 ± 8.13	10.7 ± 14.6	12.1 ± 13.51	7.66 ± 6.67	11.1 ± 16.0	7.04 ± 5.37	6.90 ± 7.69	F = 2.1 *p* = 0.903 *
	After	7.55 ± 5.44	7.45 ± 3.37	6.09 ± 2.83	8.52 ± 7.27	6.53 ± 7.31	7.10 ± 4.79	7.13 ± 7.75	
	Sig.	0.893	0.191	0.071	0.705	0.442	0.570	0.790	
ESR (mm/h)	Before	15.9 ± 11.97	14.3 ± 11.8	17.6 ± 17.6	18.45 ± 17.8	14.5 ± 13.4	16.3 ± 12.9	23.1 ± 25.1	F = 0.55 *p* = 0.755 *
	After	16.2 ± 18.7	16.3 ± 16.8	17.4 ± 19.0	20.2 ± 21.0	17.0 ± 16.2	20.2 ± 20.6	24.6 ± 25.04	
	Sig.	0.694	0.438	0.768	0.999	0.612	0.106	0.463	
WBC (×10^3^/µL)	Before	4.81 ± 1.72	4.75 ± 1.37	4.67 ± 1.31	4.90 ± 1.45	5.3 ± 3.0	5.13 ± 1.80	4.54 ± 1.56	F = 1.12 *p* = 0.996 *
	After	6.92 ± 2.21	6.78 ± 1.79	6.56 ± 1.97	6.64 ± 1.95	6.72 ± 2.83	6.91 ± 2.02	6.74 ± 2.37	
	Sig.	0.005 **	0.001 **	0.005 **	0.001 **	0.104	0.001 **	0.001 **	

* Nonsignificant difference between groups for post-test observation using ANOVA at *p* < 0.05. ** Significant at *p* < 0.05 between before and after observed values using the paired sample *t*-test.

**Table 4 ijerph-19-11798-t004:** Comparisons of CD profile.

CD Profile		Before vs. after Treatment (Mean ± S.D)							Post-Test
		Group 1 (n = 11)	Group 2 (n = 15)	Group 3 (n = 13)	Group 4 (n = 18)	Group 5 (n = 12)	Group 6 (n = 15)	Group-7 (n = 13)	Sig.
CD3	Before	72.57 ± 5.57	73.98 ± 9.31	75.83 ± 3.0	73.85 ± 7.53	78.59 ± 4.47	75.66 ± 5.29	71.99 ± 8.50	0.461
	After	72.15 ± 5.16	74.95 ± 8.23	76.02 ± 4.87	74.20 ± 7.44	76.93 ± 4.77	77.18 ± 5.59	73.78 ± 8.06	
	Sig.	0.657	0.609	0.861	0.777	0.117	0.638	0.279	
CD19	Before	9.54 ± 2.81	8.81 ± 3.83	9.11 ± 4.72	9.05 ± 4.01	10.38 ± 3.48	8.31 ± 2.89	6.89 ± 3.69	0.295
	After	14.31 ± 3.37 *	11.65 ± 4.67 *	11.04 ± 4.34 *	12.95 ± 5.04 *	13.55 ± 4.78 *	11.63 ± 3.84 *	10.50 ± 4.38 *	
	Sig.	0.010	0.001	0.039	0.000	0.003	0.005	0.005	
CD16+/CD56	Before	17.23 ± 4.42	16.47 ± 9.13	13.99 ± 3.69	16.11 ± 6.12	9.93 ± 4.08	14.12 ± 6.01	19.81 ± 9.36	0.234
	After	12.44 ± 5.45 *	12.09 ± 7.04 *	11.71 ± 5.48	11.72 ± 4.36 *	8.17 ± 3.76	10.19 ± 4.68 *	14.48 ± 9.29 *	
	Sig.	0.008	0.005	0.055	0.001	0.136	0.001	0.001	
CD3+/CD4+	Before	43.27 ± 6.01	41.75 ± 9.74	40.00 ± 7.24	42.67 ± 8.35	46.21 ± 8.33	47.12 ± 4.98	39.43 ± 8.33	0.136
	After	44.45 ± 6.76	43.61 ± 9.43	38.17 ± 12.13	44.25 ± 7.07	45.49 ± 7.52	47.91 ± 5.49	42.08 ± 10.51 *	
	Sig.	0.286	0.173	0.552	0.306	0.480	0.691	0.033	
CD3+/CD8+	Before	26.79 ± 4.87	30.49 ± 6.98	33.55 ± 7.89	28.24 ± 6.70	29.45 ± 9.89	29.35 ± 7.83	30.12 ± 7.77	0.340
	After	25.15 ± 5.07	29.50 ± 6.43	31.74 ± 7.17	26.56 ± 8.42	29.34 ± 8.09	29.23 ± 7.53	29.61 ± 7.38	
	Sig.	0.062	0.233	0.173	0.528	0.814	0.865	0.552	
T-HELPER/T-SUPPRESSOR RATIO	Before	1.68 ± 0.45	1.48 ± 0.65	1.30 ± 0.48	1.62 ± 0.60	1.41 ± 0.30	1.78 ± 0.76	1.43 ± 0.57	0.591
	After	1.86 ± 0.55	1.59 ± 0.70	1.39 ± 0.46	1.52 ± 0.54	1.90 ± 1.56	1.78 ± 0.65	1.54 ± 0.59	
	Sig.	0.103	0.088	0.328	0.616	0.638	0.910	0.064	
CD3 Abs cnt	Before	1534.58 ± 684.36	1495.31 ± 1052.90	1798.11 ± 596.49	1642.30 ± 809.47	1806.99 ± 1137.98	1398.58 ± 881.03	1272.97 ± 705.45	0.551
	After	1913.36 ± 569.83	1888.60 ± 754.61 *	1887.65 ± 645.65	1828.90 ± 607.53	2116.15 ± 756.48	1874.67 ± 788.32 *	1517.86 ± 685.32	
	Sig.	0.248	0.023	0.552	0.170	0.117	0.003	0.196	
CD19 Abs cnt	Before	212.07 ± 130.31	164.01 ± 120.10	250.80 ± 240.22	196.62 ± 128.40	246.87 ± 183.70	153.03 ± 108.16	105.46 ± 58.76	0.221
	After	393.75 ± 192.27 *	278.77 ± 144.68 *	295.66 ± 219.53	344.79 ± 225.68 *	381.60 ± 206.75 *	295.46 ± 190.84 *	212.17 ± 116.04 *	
	Sig.	0.041	0.001	0.116	0.006	0.028	0.004	0.002	
CD16+ CD56 Abs cnt	Before	345.33 ± 138.71	323.59 ± 296.55	329.54 ± 109.72	332.57 ± 163.42	209.33 ± 117.18	208.18 ± 102.09	301.24 ± 213.18	0.607
	After	309.38 ± 108.03	290.89 ± 196.13	265.35 ± 87.81	290.14 ± 138.16	213.56 ± 105.62	223.77 ± 113.88	279.69 ± 233.50	
	Sig.	0.424	0.394	0.028	0.267	0.875	0.532	0.382	
CD3+/CD4+ Abs cnt	Before	931.99 ± 479.65	728.31 ± 334.49	1017.77 ± 475.71	864.17 ± 303.56	1028.44 ± 578.31	846.13 ± 480.68	633.83 ± 328.46	0.468
	After	1200.94 ± 468.37	1012.03 ± 333.43 *	1042.84 ± 437.75	1115.23 ± 448.10 *	1201.43 ± 323.73	1160.51 ± 485.67 *	889.63 ± 424.33 *	
	Sig.	0.248	0.002	0.972	0.018	0.084	0.005	0.019	
CD3+/CD8+ Abs cnt	Before	548.44 ± 218.77	546.30 ± 297.28	796.24 ± 222.05	576.95 ± 244.41	701.23 ± 549.44	545.62 ± 408.56	450.82 ± 202.27	0.466
	After	647.00 ± 144.35	690.14 ± 234.42 *	769.13 ± 239.94	658.75 ± 158.52	820.91 ± 434.80	712.76 ± 384.16 *	590.42 ± 289.93 *	
	Sig.	0.213	0.012	0.422	0.170	0.099	0.006	0.033	

Nonsignificant difference between groups for post-test observation using ANOVA at *p* < 0.05. * Significant at *p* < 0.05 between before and after observed values using the paired sample *t*-test.

## Data Availability

The RCT protocol and all data sets analyzed during the current study are available from the corresponding author upon reasonable request. Due to data protection restrictions and participant confidentiality, we do not make participant data publicly available.

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
