# Peer review of "The Effect of Short Treatment with Nigella Sativa on Symptoms, the Cluster of Differentiation (CD) Profile, and Inflammatory Markers in Mild COVID-19 Patients: A Randomized, Double-Blind Controlled Trial"

_ijerph, 2022, doi:10.3390/ijerph191811798_

Round 1
Reviewer 1 Report
Language need to be improved significantly, Control group was not described properly, COVID test report of the patients were not supplied, Inflammatory data need to be included separately, Importance of Nigella Sativa seed is lacking,
Author Response
Reviewer-1
Open Review
( ) I would not like to sign my review report
(x) I would like to sign my review report
English language and style
(x) Extensive editing of English language and style required
( ) Moderate English changes required
( ) English language and style are fine/minor spell check required
( ) I don't feel qualified to judge about the English language and style
|
Yes |
Can be improved |
Must be improved |
Not applicable |
|
|
Does the introduction provide sufficient background and include all relevant references? |
( ) |
(x) |
( ) |
( ) |
|
Are all the cited references relevant to the research? |
( ) |
(x) |
( ) |
( ) |
|
Is the research design appropriate? |
( ) |
( ) |
(x) |
( ) |
|
Are the methods adequately described? |
( ) |
(x) |
( ) |
( ) |
|
Are the results clearly presented? |
( ) |
( ) |
(x) |
( ) |
|
Are the conclusions supported by the results? |
( ) |
( ) |
(x) |
( ) |
Comments and Suggestions for Authors
*Language need to be improved significantly,
Author Response:
Thank you very much for your comment. The English editing and proofreading have been done. Please see the attached English editing certificate.
*The Control group was not described properly,
Author Response:
Thank you very much for your comment. The control group has been clearly described in the methods section lines 154-157.
*COVID test reports of the patients were not supplied,
Author Response:
Thank you very much for your comment. PCR test was performed before and after interventions. The PCR test was positive for all patients on day one of the clinical trial. However, all PCT tests were negative after ten days of the treatment. Please see lines 149 in the methods section and 220 in the results section.
*Inflammatory data need to be included separately,
Author Response:
*The importance of Nigella Sativa seed is lacking,
Author Response:
Submission Date
07 August 2022
Date of this review
13 Aug 2022 05:56:03
End of the response to the editor and reviewer’s report

Reviewer 2 Report
The effort of the authors is appreciated, as the topic seems to be relevant and promising. Asymptomatic and mild cases of COVID-19 occur as an acute self-limiting viral infection. Also, most studies reporting the effect of Nigella Sativa have a long treatment duration of at least 4 weeks. Assessing the effect of NS in a study of shorter duration and small sample size might produce snags. The article requires some major revisions.
*Remove the word (The introduction) in 1st sentence of the introduction section.
*Mention the relevance of using Nigella sativa (NS) as a whole seed and powder as a different arm in the same study.
Methodology:
*Calculation of sample size by comparing Hydroxychloroquine improvement to NS appears impractical.
*Provide a rationale for using so many groups for the intervention drugs (one capsule, three capsules, five, and so on). Was this study based on a literature review or any previous trial suggesting this type of dosage? Are the authors trying to find the effective dosage of NS in COVID-19 in humans? Explain.
*A CONSORT diagram should be attached revealing the number of patients in inclusion, randomization, loss to follow-up, and analysis.
*This study is trying to define several questions in one trial. The primary outcome is not defined. Appropriate p-value adjustments in clinical trials should be done when multiple outcome measures are used.
*The treatment period was reduced from 4 weeks to just 10 days. Describe whether this protocol amendment is registered or not.
*Add reference to line numbers 148-150 in page 4.
Result:
*Authors have mentioned that the number of subjects included in the final analysis was 262. From figure 1, it appears that some patients have a loss to follow up at the end of the study. CD profile assessment was done for 97 patients only. Explain the handling of missing data. Also describe whether this will affect the final result of the study.
Author Response
Reviewer-2
Open Review
( ) I would not like to sign my review report
(x) I would like to sign my review report
English language and style
( ) Extensive editing of English language and style required
( ) Moderate English changes required
(x) English language and style are fine/minor spell check required
( ) I don't feel qualified to judge about the English language and style
|
Yes |
Can be improved |
Must be improved |
Not applicable |
|
|
Does the introduction provide sufficient background and include all relevant references? |
( ) |
(x) |
( ) |
( ) |
|
Are all the cited references relevant to the research? |
(x) |
( ) |
( ) |
( ) |
|
Is the research design appropriate? |
( ) |
(x) |
( ) |
( ) |
|
Are the methods adequately described? |
( ) |
(x) |
( ) |
( ) |
|
Are the results clearly presented? |
( ) |
(x) |
( ) |
( ) |
|
Are the conclusions supported by the results? |
(x) |
( ) |
( ) |
( ) |
Comments and Suggestions for Authors
The effort of the authors is appreciated, as the topic seems to be relevant and promising. Asymptomatic and mild cases of COVID-19 occur as an acute self-limiting viral infection. Also, most studies reporting the effect of Nigella Sativa have a long treatment duration of at least 4 weeks. Assessing the effect of NS in a study of shorter duration and small sample size might produce snags. The article requires some major revisions.
*Remove the word (The introduction) in 1st sentence of the introduction section.
Author Response:
Thank you very much for your comment. The word (The introduction) has been removed from the first sentence of the introduction section. Please see line 37.
*Mention the relevance of using Nigella sativa (NS) as a whole seed and powder as a different arm in the same study.
Author Response:
Thank you very much for your comment. We wanted to investigate whether the grinding of Nigella sativa has any effect on the potency of the seeds to fight the infection. If the study has shown an effect of the seed, then we could conclude which form of Nigella sativa was more potent, the whole seed or the grinded seed.
Methodology:
*Calculation of sample size by comparing Hydroxychloroquine improvement to NS appears impractical.
Author Response:
Thank you very much for your comment. Sample size planning for a clinical study is based on an estimate from previous information, which may be of different precision in different studies. We calculated our sample size based on previous research, which considered the effect of hydroxychloroquine on COVID-2 SARs. We put our hypothesis that 70% of patients who receive NS will be cleared of the virus within ten days and tested such a hypothesis. Unlike the statistical power and significance level, which are generally chosen by convention, the underlying expected event rate (in the standard or control group) must be established by other means, usually from previous studies, including observational cohorts. These often provide the best information available. It is well known that hydroxychloroquine has good results in treating COVID-19 patients. Many other types of research also confirmed NS's good results for improving signs and symptoms of COVID-19 patients. We used a power of 80% to calculate our sample size; the greater the power, the confidence that a significant result will be detected, and the greater the necessary sample size for the study. We have mentioned that the small sample size is one of the study's limitations that should be considered when interpreting the study results.
*Provide a rationale for using so many groups for the intervention drugs (one capsule, three capsules, five, and so on). Was this study based on a literature review or any previous trial suggesting this type of dosage? Are the authors trying to find the effective dosage of NS in COVID-19 in humans? Explain.
Author Response:
*A CONSORT diagram should be attached revealing the number of patients in inclusion, randomization, loss to follow-up, and analysis.
Author Response:
Thank you very much for your comment. The consort chart was designed and inserted in the results section. Please see lines 184-188.
*This study is trying to define several questions in one trial. The primary outcome is not defined. Appropriate p-value adjustments in clinical trials should be done when multiple outcome measures are used.
Author Response:
*The treatment period was reduced from 4 weeks to just 10 days. Describe whether this protocol amendment is registered or not.
Author Response:
Thank you very much for your comment. This amendment was not reflected in the registration of the trial as it was a decision of the ministry of health authorities that was mandatory to follow.
*Add reference to line numbers 148-150 on page 4.
Author Response:
Result:
*Authors have mentioned that the number of subjects included in the final analysis was 262. From figure 1, it appears that some patients have a loss to follow up at the end of the study. CD profile assessment was done for 97 patients only. Explain the handling of missing data. Also, describe whether this will affect the final result of the study.
Author Response:
Submission Date
07 August 2022
Date of this review
22 Aug 2022 07:49:30
End of the response to the editor and reviewer’s report

Reviewer 3 Report
I read the paper carefully and realized that it needs a major revision. The title of this paper sounds good, However, some ambiguous points within this submission should be addressed, modified, or clarified in its revised form.
My general remarks are as follows:
1. The English style of your manuscript needs major corrections. However, in some minor cases, some parts should be reorganized once again.
2. Throughout the manuscript, the Latin scientific names must be in Italic manner
3. In the abstract section, kindly avoid the use of abbreviations as CD profile.
4. Kindly revise the citation style of IJERPH
5. I could not find this clinical trial registration and I found only the IRB registration. However, this work needs to register before starting their investigation “The registration of all interventional trials is considered to be a scientific, ethical and moral responsibility because There is a need to ensure that decisions about health care are informed by all of the available evidence”. Kindly read the WHO website regarding this issue
https://www.who.int/clinical-trials-registry-platform/network/trial-registration#:~:text=Why%20is%20trial%20registration%20Important,all%20of%20the%20available%20evidence
6. Uniformity in the listed references must be as per journal guidelines. For further assistance, the authors can use the general guidelines of the journal or its sample template from the relevant official website.
Author Response
Reviewer-3
Open Review
( ) I would not like to sign my review report
(x) I would like to sign my review report
English language and style
(x) Extensive editing of English language and style required
( ) Moderate English changes required
( ) English language and style are fine/minor spell check required
( ) I don't feel qualified to judge about the English language and style
|
Yes |
Can be improved |
Must be improved |
Not applicable |
|
|
Does the introduction provide sufficient background and include all relevant references? |
( ) |
(x) |
( ) |
( ) |
|
Are all the cited references relevant to the research? |
( ) |
( ) |
(x) |
( ) |
|
Is the research design appropriate? |
(x) |
( ) |
( ) |
( ) |
|
Are the methods adequately described? |
(x) |
( ) |
( ) |
( ) |
|
Are the results clearly presented? |
(x) |
( ) |
( ) |
( ) |
|
Are the conclusions supported by the results? |
(x) |
( ) |
( ) |
( ) |
Comments and Suggestions for Authors
I read the paper carefully and realized that it needs a major revision. The title of this paper sounds good, However, some ambiguous points within this submission should be addressed, modified, or clarified in its revised form.
My general remarks are as follows:
The English style of your manuscript needs major corrections. However, in some minor cases, some parts should be reorganized once again.
Author Response:
Thank you very much for your comment. The English editing and proofreading have been done. Please see the attached English editing certificate.
- Throughout the manuscript, the Latin scientific names must be in the Italic manner
Author Response:
- In the abstract section, kindly avoid the use of abbreviations as CD profile.
Author Response:
Thank you very much for your comment. Abbreviations Have been changed into full names in the abstract.
- Kindly revise the citation style of IJERPH
Author response:
Thank you very much for your comment. The citation style of IJERPH has been followed throughout the manuscript.
- I could not find this clinical trial registration and I found only the IRB registration. However, this work needs to register before starting its investigation “The registration of all interventional trials is considered to be a scientific, ethical and moral responsibility because There is a need to ensure that decisions about health care are informed by all of the available evidence”. Kindly read the WHO website regarding this issue
https://www.who.int/clinical-trials-registry-platform/network/trial-registration#:~:text=Why%20is%20trial%20registration%20Important,all%20of%20the%20available%20evidence
Author Response:
Thank you very much for your comment. The trial has been registered in the Saudi Food and Drugs Authority number dated as stated in line 366.
- Uniformity in the listed references must be as per journal guidelines. For further assistance, the authors can use the general guidelines of the journal or its sample template from the relevant official website.
Author Response:
Thank you very much for your comment. The reference style of IJERPH has been followed.
Submission Date
07 August 2022
Date of this review
20 Aug 2022 07:37:27
End of the response to the editor and reviewer’s report

Round 2
Reviewer 1 Report
Acceptable for publication
Reviewer 3 Report
The authors established all the required corrections